# Oxidative Stress Produced by Paraquat Reduces Nitrogen Fixation in Soybean-*Bradyrhizobium diazoefficiens* Symbiosis by Decreasing Nodule Functionality

Germán Tortosa *[iD], Sergio Parejo, Juan J. Cabrera [iD], Eulogio J. Bedmar [iD] and Socorro Mesa *[iD]

Department of Soil Microbiology and Symbiotic Systems, Estación Experimental del Zaidín, CSIC, Profesor Albareda, 1, 18008 Granada, Spain; sergio.parejo@eez.csic.es (S.P.); juan.cabrera@eez.csic.es (J.J.C.); eulogio.bedmar@eez.csic.es (E.J.B.)
* Correspondence: german.tortosa@eez.csic.es (G.T.); socorro.mesa@eez.csic.es (S.M.)

**Abstract:** Soybean (*Glycine max.*) is one of the most important legumes cultivated worldwide. Its productivity can be altered by some biotic and abiotic stresses like global warming, soil metal pollution or over-application of herbicides like paraquat (1,1′-dimethyl-4,4′-bipyridinium dichloride). In this study, the effect of oxidative stress produced by paraquat addition (0, 20, 50 and 100 μM) during plant growth on symbiotic nitrogen fixation (SNF) and functionality of *Bradyrhizobium diazoefficiens*-elicited soybean nodules were evaluated. Results showed that the 50 μM was the threshold that *B. diazoefficiens* can tolerate under free-living conditions. In symbiosis with soybean, the paraquat addition statistically reduced the shoot and root dry weight of soybean plants, and number and development of the nodules. SNF was negatively affected by paraquat, which reduced total nitrogen content and fixed nitrogen close to 50% when 100 μM was added. These effects were due to the impairment of nodule functionality and the increased oxidative status of the nodules, as revealed by the lower leghaemoglobin content and the higher lipid peroxidation in soybean nodules from paraquat-treated plants.

**Keywords:** bacteroid; fixed nitrogen; leghaemoglobin; lipid peroxidation; nitrogen content; nitrogen fixation; oxidative stress; paraquat; soybean; symbiosis

## 1. Introduction

Legumes are one of the most important world-wide crops for humans, besides Gramineae [1]. Legumes provide several products based on plant proteins for animal and human food [2], as well as benefits for the environment, farmers, food security and health for humans and livestock [3]. Also, legumes have a positive influence on the environment and climate change, by favouring sustainable cropping systems, increasing soil fertility and reducing greenhouse gas emissions when intercropping and rotation with cereals or other crops are performed [3,4]. Indeed, grain legumes like *Trigonella*, *Lathyrus*, *Phaseolus*, *Vicia*, *Pisum*, *Lathyrus*, *Lupinus* or *Glycine*, which are commonly cultivated in Oceania, North and South America, Africa, Asia and Europe, have important grain and aerial nitrogen concentrations to address these issues [5]. Concretely, soybean (*Glycine max* L.) together with wheat, maize and rice, is one of the main crops for food production, reaching crop yields of 3.1, 4.0, 8.9 and 5.0 tonnes ha$^{-1}$ within 2019, respectively [6]. Brazil, the United States, Argentina and China, with 37%, 28%, 16% and 5% of the world's production, respectively, are the main producers of soybeans [7].

It is well known that legumes can establish a beneficial symbiosis with several soil bacteria by forming root nodules, a specific organ in which symbiotic nitrogen fixation (SNF) takes place [8–11]. This process is essential for the environment and the biosphere, coming just after photosynthesis in importance [12], and it is mainly carried out by prokaryotes which have the enzyme nitrogenase. This enzyme catalyses the reduction

of atmospheric dinitrogen ($N_2$) into an easily-available nitrogen ($NH_3$) for plants, during a highly cost-effective process carried out under low-oxygen conditions maintained by legheamogloblins (Lb) within the nodule, among other factors [13]. This specific interaction has been studied for decades by scientists, involving chemical, biochemical and molecular approaches [14,15].

Besides its agronomical and environmental relevance to the biospheric nitrogen cycle compared to synthetic N fertilisers [16], SNF can be altered by several biotic and abiotic factors like pests and pathogens, high temperature, drought and soil acidity [17]. For the latter, herbicides like paraquat (1,1´-dimethyl-4,4´-bipyridinium dichloride, also known as methyl viologen), have been widely applied in soybean crops for weed management [18,19]. Paraquat is a fast-acting and non-selective herbicide, which has a strong oxidative effect on cells by catalysing reactive oxygen species (ROS) formation like superoxide free radicals [20,21]. Concerning SNF, it has been known that paraquat addition can negatively affect $N_2$ fixation and soybean nodulation, but its effects can be site-specific depending on soil, climate conditions or even plant cultivars [18] and references therein [22].

Taking into account all these issues, the aim of this research was to evaluate the effect of paraquat addition on the symbiosis between the soybean plant and its natural endosymbiont, *Bradyrhizobium diazoefficiens*, and more specifically, on the metabolic functionality of soybean root nodules. To achieve this objective, plant physiology, nodulation, $N_2$ fixation, oxidative stress and Lb content in soybean root nodules were evaluated after paraquat addition (0, 20, 50 and 100 μM) during soybean plants' growth.

## 2. Materials and Methods

### 2.1. Bacterial Culture

*B. diazoefficiens* 110 *spc*4 [23,24] was used as soybean inoculant. This strain is a spontaneous spectinomycin-resistant derivative of *B. diazoefficiens* USDA 110 (United States Department of Agriculture, Beltsville, MD). *B. diazoefficiens* 110 *spc*4 was routinely grown at 30 °C using a modified peptone-salts-yeast extract (PSY) medium [25] (Table 1) in both Petri dishes (solid culture) and autoclaved Erlenmeyer flasks (liquid culture) by orbital shaking (170 rpm).

**Table 1.** Modified peptone-salts-yeast extract (PSY) medium composition used for *B. diazoefficiens* 110 *spc*4 growth.

| Composition [1] | Concentration |
|:---:|:---:|
| | g $L^{-1}$ |
| $KH_2PO_4$ | 0.3 |
| $K_2HPO_4$ | 0.3 |
| $MgSO_4 \times 7H_2O$ | 0.1 |
| Peptone | 3 |
| Yeast extract | 1 |
| | mg $L^{-1}$ |
| $CaCl_2 \times H_2O$ | 5 |
| $Na_2MoO_4 \times 2H_2O$ | 0.1 |
| $H_3BO_3$ | 10 |
| $ZnSO_4 \times 7H_2O$ | 1 |
| $CuSO_4 \times 5H_2O$ | 0.5 |
| $FeCl_3$ | 1 |
| $MnCl_2 \times 6H_2O$ | 0.5 |
| Spectinomycin [2] | 100 μg $mL^{-1}$ |
| Arabinose [2] | 0.1% (*w/v*) |
| Agar [3] | 15 g $L^{-1}$ |

[1] Adjusted to pH 7 with NaOH 1N. [2] Added after autoclaving. [3] Only for solid medium.

The effect of paraquat on *B. diazoefficiens* 110 *spc*4 was performed by measuring the evolution of the optical density at $\lambda$ = 600 nm ($OD_{600\,nm}$) during growth. Bacteria were incubated in a flat-bottom 96-well cell culture plate, where 100 µL of bacterial suspension ($OD_{600\,nm}$ = 0.05), 100 µL of the corresponding paraquat solution in PSY and 30 µL of mineral oil (Merck, M8410) were added to each well. The cell culture plate was incubated in darkness at 30 °C without shaking, and the $OD_{600\,nm}$ was periodically measured using a SunriseTM Absorbance Microplate Reader (Tecan Trading AG, Mannedorf, Switzerland). Two sets of paraquat concentrations, at a low range (0, 10, 20, 30, 40 and 50 µM), and a high range (0, 100, 200, 300, 400 and 500 µM) were assayed. Data were expressed as the mean of 5 replicates.

For plant experiments, bacterial inoculant was prepared as follows: *B. diazoefficiens* 110 *spc*4 was grown over 5–6 days until stationary phase was reach ($OD_{600\,nm}$ = 1). Then, bacterial culture was centrifuged at 7500× *g* at 4 °C for 10 min and subsequently, resuspended in sterilised saline solution (0.9% *w/v* NaCl) at $OD_{600\,nm}$ = 0.8.

## 2.2. Plant Inoculation and Growth

Surface-sterilisation of soybean (*Glycine max* L. Merr., cv. Williams) seeds was undertaken as previously described [26]. Briefly, soybean seeds were consecutively submerged into 96% (*v/v*) ethanol and 30% (*v/v*) $H_2O_2$ for 5 and 15 min, respectively. After that, seeds were washed with sterilised distiller water and germinated into Petri dishes containing 25 mL of 1% (*w/v*) agar during 3 days at 28 °C in darkness.

Plants were grown in 0.25 L pots containing vermiculite N° 3 (1–4 mm, 80–100 kg $m^{-3}$) as growing substrate, and assembled into Leonard jars [27]. One germinated seed was transferred into each Leonard jar, followed by inoculation with 1 mL of bacterial inoculant ($OD_{600\,nm}$ = 0.8, $\approx 10^8$ cells $mL^{-1}$). Leonard jars were watered twice per week with 200 mL of a modified Jensen mineral solution [28] (Table 2), supplemented or not supplemented with the corresponding paraquat concentration (0, 20, 50 and 100 µM). Plants were grown for 28 days post inoculation until harvest at V3-V4 vegetative stage [29] (Figure 1). The growing condition were day/night temperatures of 26–22 °C, 16–8 h day/night cycle and photosynthesis photon flux density of 180 µmol photons $m^{-2}$ $s^{-1}$, respectively [26].

A detailed description about experimental design including replication is shown in Section 2.4.

**Table 2.** Modified Jensen mineral solution composition used for soybean plant experiments.

| Composition [1] | Concentration |
|---|---|
| | g $L^{-1}$ |
| $CaHPO_4$ | 0.42 |
| $CaSO_4 \times 2H_2O$ | 0.54 |
| $K_2HPO_4$ | 0.08 |
| $MgSO_4 \times 7H_2O$ | 0.08 |
| NaCl | 0.08 |
| $FeCl_3 \times 6H_2O$ | 0.07 |
| | mg $L^{-1}$ |
| $MnCl_2 \times 6H_2O$ | 0.21 |
| $Na_2MoO_4 \times 2H_2O$ | 0.04 |
| $H_3BO_3$ | 4.23 |
| $ZnSO_4 \times 7H_2O$ | 0.42 |
| $CuSO_4 \times 5H_2O$ | 0.21 |

[1] Adjusted to pH 7 with 1 N NaOH.

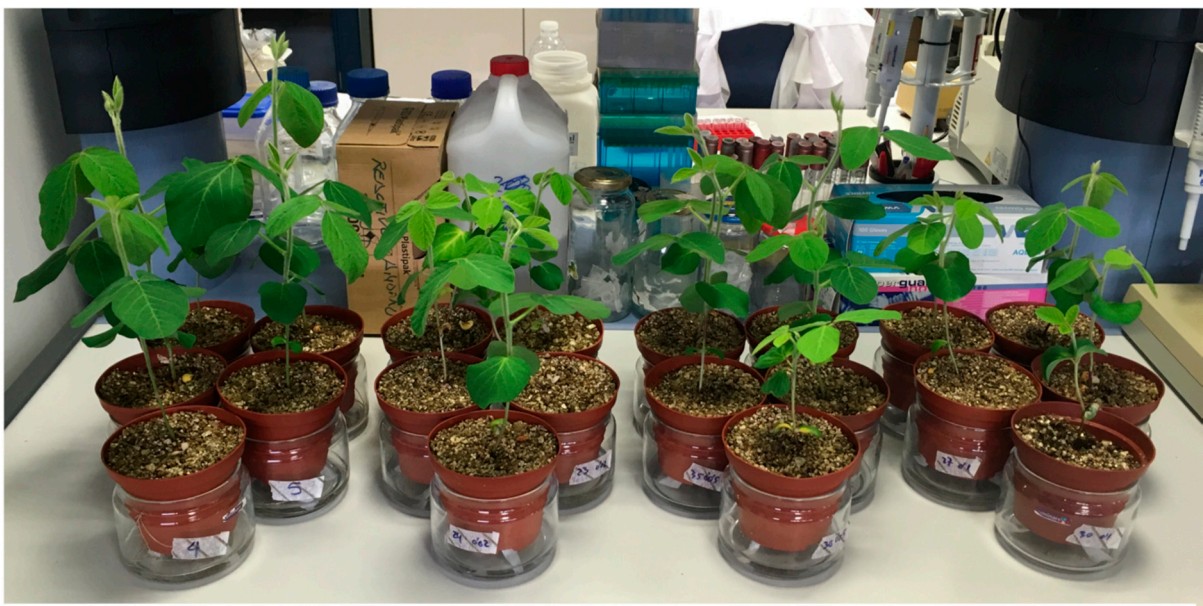

**Figure 1.** Leonard jars used for growth of soybean plants inoculated with *B. diazoefficiens* 110 *spc*4 and treated or not treated with paraquat. From left to right: 0, 20, 50 and 100 μM, respectively.

### 2.3. Analyses

Plant physiology parameters like shoot and root dry weight (SDW and RDW), nodule number (NN) and nodule fresh and dry weight (NFW and NDW) per plant were measured after harvesting as described in reference [26]. SDW, RDW, NDW and soybean seeds were weighed after 3 days at 70 °C, which were ground to less than 0.5 mm for nitrogen (N) determination. Also, a representative number of fresh nodules were frozen in liquid nitrogen and stored at −80 °C for further biochemical assays.

N content in SDW ($N_{shoot}$), RDW ($N_{root}$) and NDW ($N_{nodules}$) were determined by the Dumas method using a LECO TruSpec CN Elemental Analyzer (Elemental Analysis Service, Estación Experimental del Zaidín, EEZ-CSIC) [26]. Also, N content in seeds ($N_{seed}$) was determined (5.39% or 6.05 mg per seed). SNF was estimated by the Total N difference method [30] as the mass balance between fixed N of plant biomass and initial N (N in seeds) as follows: Fixed nitrogen (FN) = ($N_{shoot}$ + $N_{root}$ + $N_{nodules}$) − $N_{seed}$

Nodular fraction was obtained following the methodology described by Tortosa and co-workers [26]: Briefly, 0.5 to 1.0 g of NFW were manually homogenised by using a cooled porcelain pestle and mortar with 6 mL of buffer solution [50 mM $Na_2HPO_4 \times 2H_2O/NaH_2PO_4 \times 2H_2O$, pH 7.4; 0.02% *w/v* $K_3Fe(CN)_6$, and 0.1% *w/v* $NaHCO_3$] and 0.1 g of polyvinyl poly(vinylpolypyrrolidone) (PVPP). After that, the extract was centrifuged at 12,000× *g* at 4 °C for 20 min.

Leghaemoglobin (Lb) content in the nodular fraction was fluorimetrically determined after an acidic reaction at 120 °C during 30 min as previously described [31]. In brief, 50 μL of nodular fraction was added to a glass tube containing 3.15 mL of oxalic acid (66 g $L^{-1}$) and autoclaved at 120 °C for 30 min. After cooling down, the fluorescence in each tube was measured by using a Shimadzu spectrophotofluorometer (Shimadzu Scientific Instruments, Kyoto, Japan) ($\lambda_{excitation}$ = 405 and $\lambda_{absorption}$ = 600 nm). Non-autoclaved tubes containing acidic nodular fraction were used as a control.

Lipid peroxidation was also determined in the nodule [32]. Nodular fraction was incubated at 100 °C with a reaction mixture containing trichloroacetic acid (TCA), thiobarbituric acid (TBA) and chlorhydric acid. After that, TBA-reacting substances (TBARS) were measured spectroscopically at λ = 535 nm and compared with malondialdehyde (MDA) as standard.

*2.4. Experimental Design and Statistical Analysis*

Plant experiments were designed according to the recommendations of Gomez and Gomez [33]. We carried out some single-factor experiments based on the randomised block procedure in order to evaluate the effect of added paraquat to plant physiology and nodular development. Reproducibility was checked by doing a total of three consecutive plant experiments, which were carried out during two years of experimentation (2017–2018). Also, repeatability was assayed by using a total of ten plants (or replicates) per treatment for each plant experiment. Finally, data were expressed as a compilation of all experiments according to their reproducibility and repeatability.

For each experiment and parameter, a descriptive statistical analysis was undertaken, including mean, absolute and relative error calculations. In order to test the effect of paraquat in bacterial and plant development, inferential statistical analyses based on the analysis of variance (ANOVA, one-way) with a post hoc Tukey test, $p < 0.05$) were calculated. These analyses were undertaken assuming normal distribution and homoscedasticity of the raw data.

All statistical analyses were carried out by using the Spanish versions of LibreOffice Calc (v6.0.7.3) (https://www.gnu.org/software/pspp/) and GNU-PSPP open-source (v1.0.1) software (https://es.libreoffice.org/descubre/calc/), respectively.

## 3. Results

*3.1. Bacterial Growth*

The effect of paraquat addition on bacterial growth is shown in Figure 2. Two sets of experiments were performed at low and at high range of paraquat concentrations. In the former, where the concentration of paraquat ranged between 0 and 50 µM (Figure 2a), *B. diazoefficiens* 110 *spc*4 grew until $OD_{600\,nm}$ 0.5 without any statistical differences either at exponential or at stationary phases. In the latter, where the concentration of paraquat added ranged between 0 and 500 µM, the bacterial growth was significantly affected ($p > 0.05$) (Figure 2b). Paraquat $\geq$100 µM substantially decreased the $OD_{600\,nm}$ from the 4th day until the end of the experiment, including early-log and stationary phases. These results meant that *B. diazoefficiens* 110 *spc*4 can properly tolerate paraquat up to 50 µM without affecting its growth and development, this concentration being the threshold that this strain can tolerate without suffering severe stress.

*3.2. Plant Physiology*

Paraquat added to soybean plants negatively affected plants growth (Table 3). SDW decreased nearly 1.2 and 1.4 fold with 20 and 50 µM treatments, respectively, compared to plants without paraquat, and halved when 100 µM was added (from 649 $\pm$ 89 to 316 $\pm$ 72 mg plant$^{-1}$). Also RDW was statistically reduced with 50 and 100 µM of paraquat, close to 1.2 and 1.6 fold in comparison to the non-treated plants. This reduction in the biomass was also found in plant development as assessed by the shoot-to-root ratio (SDW/RDW), which also decreased with the increasing paraquat concentrations added to plants. Paraquat affected nodulation by halving NN and NFW values at 100 µM in comparison to those when no paraquat was added (from 49 $\pm$ 12 to 28 $\pm$ 9, and from 366 $\pm$ 98 to 175 $\pm$ 65 mg plant$^{-1}$, respectively).

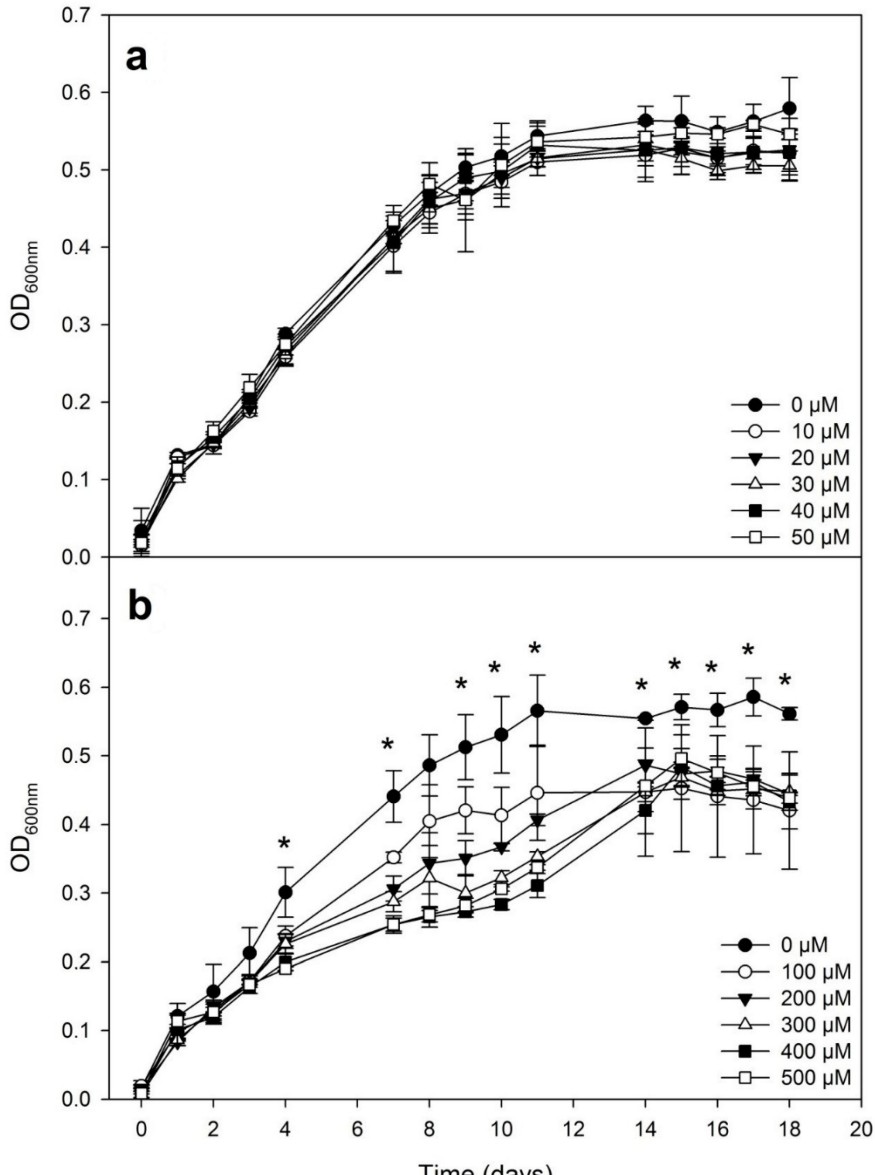

**Figure 2.** Effect of paraquat addition on *B. diazoefficiens* 110 *spc*4 growth determined by $OD_{600 \, nm}$. Paraquat was added within two ranges of concentration: 0–50 μM (**a**) and 0–500 μM (**b**). Data are expressed as the mean and standard deviation of 5 replicates. At each time, asterisk (*) means statistical differences according to the Wilcoxon–Mann–Whitney test ($p \leq 0.05$).

**Table 3.** Effect of paraquat addition on shoot dry weight (SDW), root dry weight (RDW), shoot-to-root ratio (SDW/RDW), nodule number (NN), nodule fresh weight (NFW) and fresh weight per nodule (NFW/NN) of soybean plants inoculated with *B. diazoefficiens* 100 *spc*4.

| Paraquat Added to Mineral Solution (μM) | SDW (mg plant$^{-1}$) | RDW (mg plant$^{-1}$) | SDW/RDW (plant$^{-1}$) | NN (plant$^{-1}$) | NFW (mg plant$^{-1}$) | NFW/NN (mg Nodule$^{-1}$) |
|---|---|---|---|---|---|---|
| 0 | 649 a | 217 a | 2.99 a | 49 a | 366 a | 7.47 a |
| 20 | 533 b | 217 a | 2.46 b | 47 a | 342 b | 7.28 a |
| 50 | 459 c | 184 b | 2.49 b | 37 a | 273 c | 7.38 ab |
| 100 | 316 d | 132 c | 2.40 b | 28 b | 175 d | 6.25 b |

Values in a column followed by the same letter are not statistically different according to Wilcoxon–Mann–Whitney test ($p \leq 0.05$).

### 3.3. N Content and Symbiotic Nitrogen Fixation (SNF)

After plant harvesting, N content was measured in shoots, roots and nodules. As shown in Table 4, the addition of paraquat statistically reduced N concentration in all plant tissues, especially in shoots. $N_{shoot}$ was halved when 20 or 50 μM paraquat were added compared to the control plants ($23.29 \pm 3.78$ mg plant$^{-1}$), and it was further reduced to 2.5 fold in comparison to the control plants when 100 μM was applied ($9.39 \pm 1.23$ mg plant$^{-1}$). A lesser reduction in $N_{root}$ was observed, with a decrease from $4.62 \pm 0.45$ to $4.47 \pm 0.91$, $3.39 \pm 0.57$ and $2.42 \pm 0.36$ mg plant$^{-1}$, respectively, in plants cultivated in the presence of 20, 50 and 100 μM of paraquat. Finally, $N_{nodules}$ also decreased with the addition of paraquat, especially when 100 μM was added, which reduced the N content to more than half in comparison to the control plants (Table 4).

**Table 4.** Effect of paraquat addition on nitrogen content in shoot ($N_{shoot}$), root ($N_{root}$) and nodules ($N_{nodules}$) of soybean plants inoculated with *B. diazoefficiens* 110 *spc*4.

| Paraquat Added to Mineral Solution (μM) | $N_{shoot}$ (mg plant$^{-1}$) | $N_{root}$ (mg plant$^{-1}$) | $N_{nodules}$ (mg plant$^{-1}$) |
|---|---|---|---|
| 0 | 23.29 a | 4.62 a | 2.4 a |
| 20 | 11.93 b | 4.47 b | 1.82 b |
| 50 | 12.18 b | 3.39 c | 2.15 b |
| 100 | 9.39 c | 2.42 d | 1.12 c |

Values in a column followed by the same letter are not statistically different according to Wilcoxon–Mann–Whitney test ($p \leq 0.05$).

SNF of soybean-*B. diazoefficiens* symbiosis was estimated by fixed nitrogen (FN) of plant biomass (Figure 3). The paraquat addition during plant growth statistically reduced FN in all treatments assayed. FN of soybean-*B. diazoefficiens* symbiosis was estimated as $25 \pm 4$ mg plant$^{-1}$. This value decreased between 1.6 and 2 fold in the presence of 20 and 50, and 100 μM paraquat.

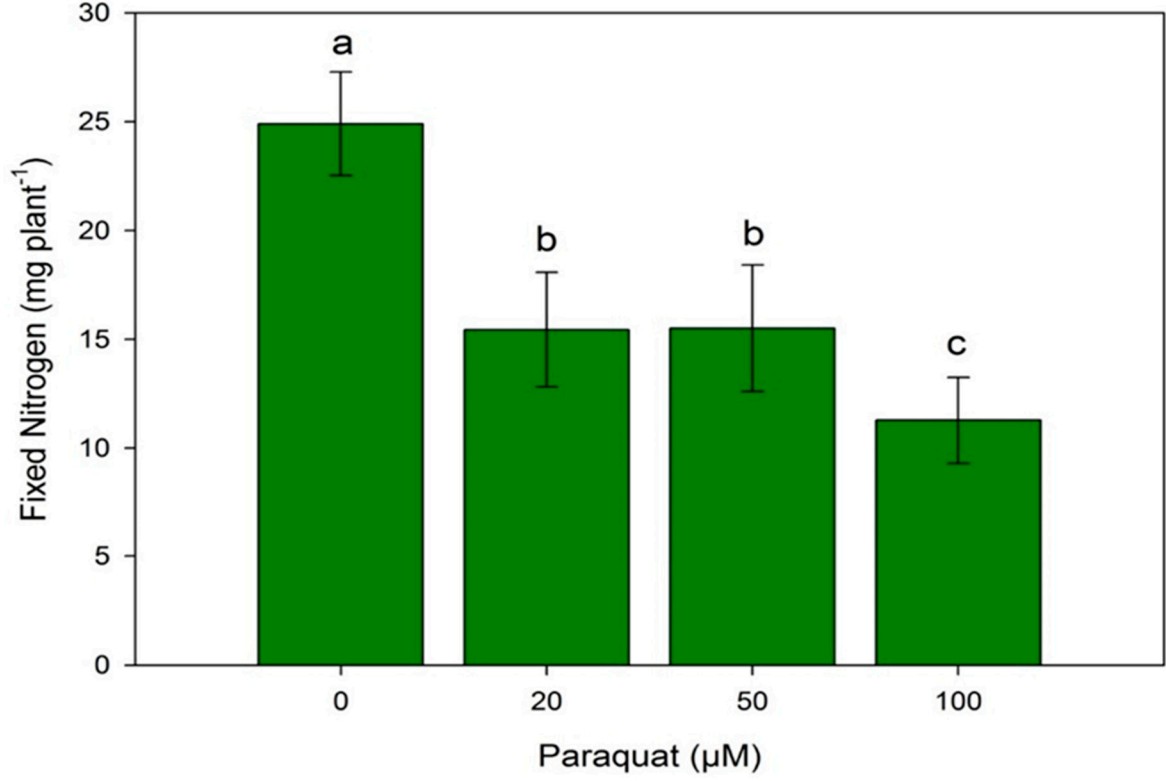

**Figure 3.** Effect of paraquat addition on fixed nitrogen (FN) by soybean plants inoculated with *B. diazoefficiens* 110 *spc*4. Similar letters are not statistically different according to the Wilcoxon–Mann–Whitney test ($p \leq 0.05$).

### 3.4. Lb Content and Lipid Peroxidation in Nodules

Physiological and oxidative status of nodule functionality were assessed by measuring total Lb content and lipid peroxidation in soybean nodules (Table 5). As expected, Lb content decreased when paraquat was added to mineral solution. Control plants presented $7.02 \pm 1.02$ mg g$^{-1}$ NFW and decreased to values ranged from $4.13 \pm 0.87$ to $4.16 \pm 0.69$ mg g$^{-1}$ NFW with 20 and 50 μM paraquat, respectively. No statistical differences between these two concentrations were found. The most noteworthy effect was found when 100 μM of paraquat was added, which halved Lb content compared to control plants.

**Table 5.** Effect of paraquat addition on leghaemoglobin (Lb) content and lipid peroxidation in nodules of soybeans plants inoculated with *B. diazoefficiens* 110 *spc*4 treated or not with paraquat.

| Paraquat Added to Mineral Solution (μM) | Lb [mg (g NFW$^{-1}$)] | Lipid Peroxidation [nmol MDA (g NFW$^{-1}$)] |
|---|---|---|
| 0 | 7.02 a | 64.2 c |
| 20 | 4.13 b | 74.8 b |
| 50 | 4.16 b | 75.2 b |
| 100 | 3.58 c | 85.4 a |

NFW: nodule fresh weight; MDA: malondialdehyde. Values in a column followed by the same letter are not statistically different according to the Wilcoxon–Mann–Whitney test ($p \leq 0.05$).

Finally, lipid peroxidation was measured in the same nodule fraction as Lb. The addition of paraquat to plants provoked a significant alteration in the oxidative status of the soybean nodules. Lipid peroxidation increased from $64.2 \pm 2.76$ to $74.8 \pm 3.16$ and $75.2 \pm 2.19$ nmol MDA g NFW$^{-1}$ with 20 and 50 μM paraquat, 100 μM being the concentration which induced the most oxidative damage to nodules ($85.4 \pm 4.21$ nmol MDA g NFW$^{-1}$).

## 4. Discussion

Depending on its cellular concentrations, ROS can act as either signalling molecules or as stress regulators for a wide range of biological mechanisms [34]. Also, several biotic and abiotic conditions generate an increase in ROS content like pathogen infections, drought, and heavy metal pollution, among others [35]. It is well-known that paraquat is a ROS-inducing agent, which can alter cell metabolism and membrane structure by increasing the net production of superoxide anion radicals and hydrogen peroxide [20]). The effect of ROS on transcriptional and physiological functionality has been studied in several bacteria like *Pseudomonas aeruginosa*, *Escherichia coli* or *B. diazoefficiens* [21,36–39]. According to Donati and colleagues [21], most of these studies were focused on the effect of oxidative stressors during short periods of exposure. In our research, it was found that *B. diazoefficiens* can properly tolerate paraquat concentrations ranging from 5 to 50 μM during growth. Also, higher concentrations were also checked (100–500 μM), being 50 μM the threshold that *B. diazoefficiens* supported without slowing its growth. These findings are in agreement with Donatti and coworkers [21], who demonstrated that 1 mM paraquat strongly affected both lag phase and generation time. Indeed, these authors concluded that *B. diazoefficiens* tolerates a lower concentration of paraquat (indeed, 100 μM) by enhancing motility, translational activity and exopolysaccharide production, as well as expressing genes related to global stress like chaperones and sigma factors.

Paraquat is a herbicide used worldwide for weed management in crops, but depending on doses applied and cultivars, can also produced a negative effect on soybean yield or plant growth [18,19,22]. In this study, a decrease in plant biomass (SDW and RDW) and nodulation (NN and NFW) were recorded when paraquat was added during plant growth; 20 μM of paraquat did not affect most of these parameters, meanwhile 100 μM halved them. Hamim and colleagues [40] found an important reduction of 35% in biomass dry weight in soybeans after 2 weeks of paraquat addition. Also, Marino et al. [41] recorded a significant decrease in SDW, RDW and NDW when 10 μM of paraquat was added to pea plants. Conversely, Kucey et al. [18] assayed different herbicides, paraquat among them,

and they found an increase in growth and yield in plants compared to control treatments, which means that paraquat effect depends on soybean cultivars, doses and the agricultural system used (greenhouse or field experiments).

SNF is negatively affected by paraquat (Figure 4). Dalton [20] studied ROS production by soybean root nodules when 0.1 mM or 1 mM of paraquat was added daily to mineral solution, and both concentrations strongly reduced nitrogenase activity. Marino et al. [41] also observed a decrease in the apparent nitrogenase activity with 0.1 and 10 μM applied to pea plants. According to our results, N content and FN were negatively affected by $\geq 20$ μM paraquat, which halved them with the highest doses used (100 μM). These findings can be explained with the reduction of Lb content in nodules, which showed a similar tendency as FN. It is known that Lb is essential to maintain the optimal $O_2$ concentration inside the nodule required for nitrogenase functionality [42]. Also, Lb content can be reduced when plants are subjected to abiotic stresses like salinity, darkness, nitrate or copper concentration [26]. We confirmed that paraquat produces an important oxidative stress inside the nodule measured by a lipid peroxidation increment, which produces a reduction in Lb content and, as a consequence, affects $N_2$ fixation of soybean–*B.diazoefficiens* symbiosis.

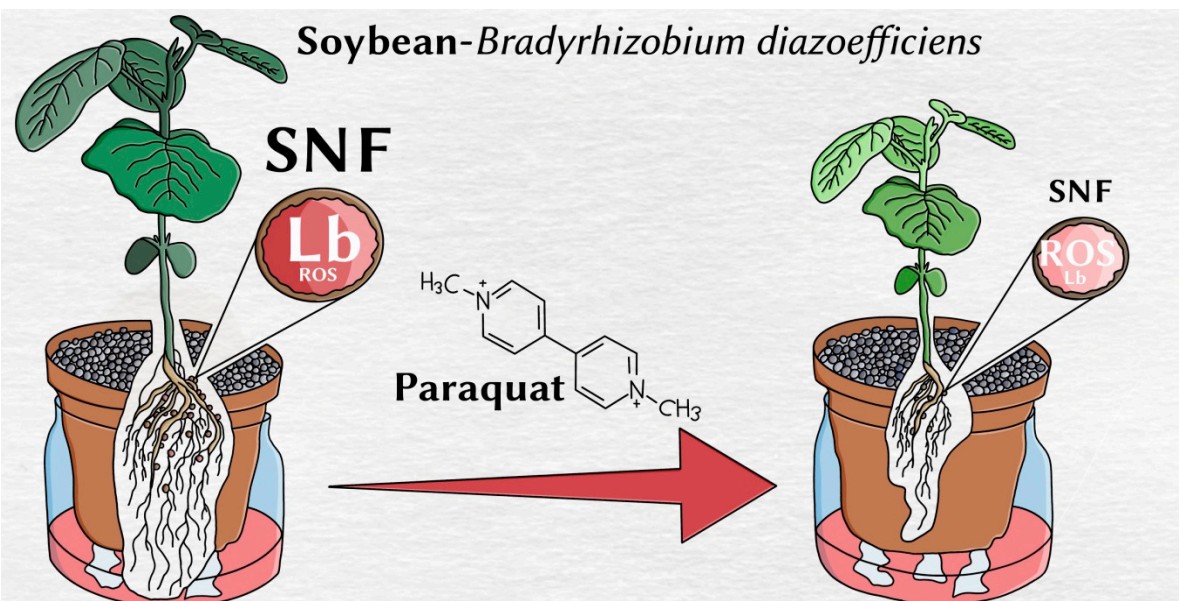

**Figure 4.** An overview of the main effect of paraquat in the soybean–*B. diazoefficiens* symbiosis. SNF: symbiotic nitrogen fixation; Lb: leghaemoglobin; ROS: reactive oxygen species.

## 5. Conclusions

In summary, our results demonstrate the paraquat addition to mineral solution negatively affected soybean growth and development by decreasing plant biomass and nodulation. Also, paraquat altered $N_2$ fixation by reducing SFN by the endosymbiont *B. diazoefficiens*. Furthermore, paraquat induced an important oxidative stress in plant cells and nodules by increasing lipid peroxidation and reducing Lb content, thus affecting nodule functionality

**Author Contributions:** G.T.: Conceptualization, Methodology, Investigation, Writing original draft, Visualization. S.P. and J.J.C.: Investigation. E.J.B.: Resources, Writing review and editing, Funding acquisition. S.M.: Conceptualization, Resources, Writing review and editing, Supervision, Project administration, Funding acquisition. All authors have read and agreed to the published version of the manuscript.

**Funding:** European Regional Development Fund (ERDF) grants from the Spanish Ministerio de Economía y Competitividad (AGL2015–63651-P, S.M.) and the Junta de Andalucía Regional Government (PE2012-AGR1968; E.J.B.) financed this work.

**Institutional Review Board Statement:** Not applicable.

**Informed Consent Statement:** Not applicable.

**Data Availability Statement:** Not applicable.

**Acknowledgments:** The authors want to thank Dulce N. Rodríguez (IFAPA, Sevilla) for providing soybean seeds and Francis Lewis for the written English revision.

**Conflicts of Interest:** The authors declare no conflict of interest. The funders had no role in the design of the study; in the collection, analyses, or interpretation of data; in the writing of the manuscript; or in the decision to publish the results.

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
