# Peer review of "Oxidative Stress Produced by Paraquat Reduces Nitrogen Fixation in Soybean-Bradyrhizobium diazoefficiens Symbiosis by Decreasing Nodule Functionality"

_nitrogen, doi:10.3390/nitrogen2010003_

Round 1

Reviewer 1 Report

The papers addresses a topic of current interest and can be accepted for publication after minor revision. Below some suggestions to be taken into consideration when revising the paper.

L21. Spell out SNF as it appears here for first time.

L.119: What was the irrigation water used to prepare nutrient solution? If it was tap water did the authors take into consideration the mineral composition of the irrigation water?

Table 2: The nutrient solution composition should be presented as mmol/L of each nutrient ion and not as g/L of each fertilizer.

Table 2: Why did the authors apply NaCl? Na is not a nutrient.

Figure 3: Statistical separation of means is missing. The SE (I suppose; it is not referenced in the Figure legend) is not sufficient.

Author Response

Reviewer 1

 Comments and Suggestions for Authors

The papers addresses a topic of current interest and can be accepted for publication after minor revision. Below some suggestions to be taken into consideration when revising the paper.

<Response>:  Thanks for your comments.

L21. Spell out SNF as it appears here for first time.

<Response>:  Thanks for your suggestion. SNF is already defined before line 21 (old version; now line 26), (see line 21). Also, SNF is defined in Introduction section (line 54).

L.119: What was the irrigation water used to prepare nutrient solution? If it was tap water did the authors take into consideration the mineral composition of the irrigation water?

<Response>: Thanks for your question. The modified Jensen solution was prepared with deionized water. We never use tap water for plant experiments in which the chemical composition could alter the treatment.

Table 2: The nutrient solution composition should be presented as mmol/L of each nutrient ion and not as g/L of each fertilizer.

<Response>: Thanks for your suggestion. However, we believe that both ways (mmol/L or g/L) are equally effective. Several arguments support our proposal to define the composition of the Jensen solution in g/L:  (i) the original recipe is expressed in that way (Vincent 1970) [28]); (ii) we have used this definition in two previous publications from our lab (Tortosa et al., 2015, https://doi.org/10.1007/s13199-015-0341-3) and Tortosa et al, 2020, (https://doi.org/10.1016/j.envexpbot.2020.104262, [26]).). Additionally, there is no requirement about that in the “Instructions for Authors” of the Nitrogen journal.

Table 2: Why did the authors apply NaCl? Na is not a nutrient.

<Response>: We agree with the reviewer. Vincent (1970) [28] optimized this recipe for nodulation and nitrogen fixation tests in legumes. NaCl was included in the Jensen solution recipe to adjust the ionic strength to similar values to bacterial inocula. In order to avoid misunderstanding we have changed de term “nutrient solution” by “mineral solution”.

Figure 3: Statistical separation of means is missing. The SE (I suppose; it is not referenced in the Figure legend) is not sufficient.

<Response>: Thanks for your comment. We forgot to add the statistical analysis in Figure 3, which is based on ANOVA one-way with post-hoc Tukey test (p<0.05) (see 2.4 section, lines 183-201). A new version of Figure 3 is included (line 303).

Reviewer 2 Report

Relatively low toxicity herbicides should be used in crops production (including soybean). Herbicides with low environmental and human risk should be selected. Paraquat, unfortunately, does not belong to this group. Many countries have banned the use of paraquat in plant cultivation. Paraquat – dichloride are classified as “highly hazardous” by PAN International.
The authors write about the pro-environmental values of the cultivation of legumes, including soybean, and at the same time test paraquat in its cultivation.
Its effects on plants, the environment and humans are well documented. It was concluded that paraquat was toxic to plants and symbiotic microorganisms. It causes nodulation failure and decreased Rhizobium population and activity.
In the assessed work was founded that paraquat negatively influenced plant growth under the conditions of the conducted experiment, which are difficult to compare with natural growing conditions. The discussion of the results confirms my fears that the conducted experiment confirms the data contained in the literature, and also presents 'the data expected'.
In the conducted study, the amount of nitrogen from biological reduction process (nitrogen fixation) has been estimated, without applying a specific, reliable method. As a result of the adopted calculation method, the obtained results may be erroneous.
In addition, the authors did not refer to the effects of less toxic herbicides than paraquat in either the research or the discussion of the results.
In the chapter 'material and methods', not specified: years in which the research was conducted; number of replications, e.g. in a vegetation experiment; variety of soybean; the data of devices used for measurements.
In the chapter 'conclusions' do not lead the discussions and inquire into the causes of the results - it should be done in the previous chapter.
The results presented may be of minor local importance in countries where paraquat is approved for use.
Considering the above, I do not recommend publishing this manuscript in the journal 'Nitrogen'

Author Response

Dear Editor,

We appreciate very much the comments and suggestions raised by the three reviewers. You will notice that we considered all of them and that almost all of the suggestions were taken into account in the revised version of the manuscript (indicated as track changes mode). Lines, figures, and references in the specific responses correspond to those of the revised manuscript. The text of the each reviewer is repeated here, and our responses are marked as <Response>.

We believe that the quality of the revised manuscript has thereby improved, and we would be happy if it is now acceptable for publication in the Nitrogen journal.

With best regards,

Germán Tortosa and Socorro Mesa

Comments and Suggestions for Authors

<Response>: First of all, we would like to thank this reviewer for his/her comments, which will help us to improve the final version of our manuscript.

Taken into account the suggestions of this reviewer, the aim of our research was probably misunderstood. We did not intend to evaluate the effect of paraquat on plant growth neither to analyze its agronomical effects using soybean-Bradyrhizobium japonicum symbiosis as a model system. Conversely, with this research we pretend to find new insights about the metabolic functionality of nodules under different levels of oxidative stress (produced by paraquat addition), with especial interest in the role of nodular legheamoglobin.

For a better understanding of our goal, we have rewritten the title, the abstract and the last paragraph of the Introduction as follows:

Title (lines 2-5): Oxidative stress produced by paraquat reduces nitrogen fixation in soybean-Bradyrhizobium diazoefficiens symbiosis by decreasing nodule functionality

Abstract (lines 13-31):

In this study, the effect of oxidative stress produced by paraquat addition (0, 20, 50 and 100 µM) during plant growth on symbiotic nitrogen fixation (SNF) and functionality of Bradyrhizobium diazoefficiens-elicited soybean nodules were evaluated.

Introduction (lines 83-89):

Taking into account all these issues, the aim of this research was to evaluate the effect of paraquat addition on the symbiosis between soybean plants and its natural endosymbiont, Bradyrhizobium diazoefficiens, and more specifically, on the metabolic functionality of soybean root nodules. To achieve this objective, plant physiology, nodulation, N2 fixation, oxidative stress and Lb content in soybean root nodules were evaluated after paraquat addition (0, 20, 50 and 100 µM) during soybean plants growth.

Relatively low toxicity herbicides should be used in crops production (including soybean). Herbicides with low environmental and human risk should be selected. Paraquat, unfortunately, does not belong to this group. Many countries have banned the use of paraquat in plant cultivation. Paraquat – dichloride are classified as “highly hazardous” by PAN International.

The authors write about the pro-environmental values of the cultivation of legumes, including soybean, and at the same time test paraquat in its cultivation.

Its effects on plants, the environment and humans are well documented. It was concluded that paraquat was toxic to plants and symbiotic microorganisms. It causes nodulation failure and decreased Rhizobium population and activity.

<Response>: Thanks for the comment. As correctly addressed by this reviewer, the negative effects of paraquat on plant cultivation and legumes symbiosis are well-documented. Indeed, some information about that was already included in the Introduction section of the original version of our manuscript (lines 69-75):

Paraquat is a fast-acting and non-selective herbicide, which has a strong oxidative effect on cells by catalysing reactive oxygen species (ROS) formation like superoxide free radicals [20-21]. Concerning to SNF, it has been known that paraquat addition can negatively affect N2 fixation and soybean nodulation, but its effects can be site-specific depending on soil, climate conditions or even plant cultivars [18] and cites therein; [22].

And also, in Discussion section (lines 352-356):

SNF is negatively affected by paraquat (Figure 4). Dalton [20] studied ROS production by soybean root nodules when 0.1 mM or 1 mM of paraquat was added daily to the mineral solution, and both concentrations strongly reduced nitrogenase activity. Marino et al [41] also observed a decrease in the apparent nitrogenase activity with 0.1 and 10 µM applied to pea plants.

In the assessed work was founded that paraquat negatively influenced plant growth under the conditions of the conducted experiment, which are difficult to compare with natural growing conditions. The discussion of the results confirms my fears that the conducted experiment confirms the data contained in the literature, and also presents 'the data expected'.

<Response>: Thanks for the observation. As shown in Figure 1, to achieve our goal (effect of paraquat-induced oxidative stress on SNF and nodule functionality), we specifically designed and performed the experiments under controlled laboratory conditions, to produce oxidative damage in the nodules. Hence, we used hydroponic jars filled with vermiculite as substrate (not soil), and paraquat was added in the mineral solution instead of foliar-spray application (its usual way). Our research focus was biological and biochemical rather than agronomic; and for that, we did not pretend to mimic natural growing conditions, which we believe that is out of the scope of this study.  

In the conducted study, the amount of nitrogen from biological reduction process (nitrogen fixation) has been estimated, without applying a specific, reliable method. As a result of the adopted calculation method, the obtained results may be erroneous.

<Response>: We are sorry to disappoint the reviewer in this comment. According to Hardarson and Danso (1993), there are several methods to evaluate nitrogen fixation in grain legumes: The dry matter methods (DM), the Total N difference method, Nodule observation, Acetylene reduction assay (ARA), Xylem-solute technique and 15N methodologies. We agree that all these methods have some advantages and disadvantages. In our research, we applied the Total N difference method mainly due to: (i) we did not use soil as a plant substrate (which allow us to properly calculate the N mass balance); (ii) N determination by Dumas’ method is well-established in our laboratory. We also evaluated SNF by using the ARA methodology; however we did not include these results in the manuscript due to the lower reliability in comparison with Total N difference method. The ARA methodology is indeed not well-accepted by the scientific community.

For a better understanding, the reference of Hardarson and Danso (1993) [30] has been added to the manuscript (line 155 in Section 2.3): Hardarson, G.; Danso, S.K.A. Methods for measuring biological nitrogen fixation in grain legumes. Plant Soil. 1993, 152, 19–23. https://doi.org/10.1007/BF00016330

In addition, the authors did not refer to the effects of less toxic herbicides than paraquat in either the research or the discussion of the results.

<Response>: Thanks for the suggestion. As we mentioned above, our research was not agronomic. Paraquat was used due to its oxidative effect on cells based on its capacity to catalyze reactive oxygen species (ROS) formation like superoxide free radicals. Actually, we are currently studying other herbicides and chemical compounds like copper to evaluate their oxidative damage in soybean nodules.

In the chapter 'material and methods', not specified: years in which the research was conducted; number of replications, e.g. in a vegetation experiment; variety of soybean; the data of devices used for measurements.

<Response>: Please, note that most of the information required by the reviewer was already shown in the original version of the article:

The number of replicates is presented in line 199-200: a total of ten plants (or replicates) per treatment for each plant experiment.

The vegetation stage is indicated in line 134: Plants were grown for 28 days post inoculation until harvest at V3-V4 vegetative stage [29] (Figure 1)

The variety of soybean is also cited in line 120: Surface-sterilisation of soybean (Glycine max L. Merr., cv. Williams).

Also, data about the devices used for measurements is already shown as follows:

  • OD600 (line 108): SunriseTM Absorbance Microplate Reader (Tecan Trading AG, Switzerland).
  • N content (line 152): LECO TruSpec CN Elemental Analyzer (Elemental Analysis Service, Estación Experimental del Zaidín, EEZ-CSIC) [26].
  • Fluorescence (lines 180-181): Shimadzu spectrophotofluorometer (Shimadzu Scientific Instruments, Kyoto, Japan).
  • The information of the rest of devices can be found in our previous article by Tortosa et al, 2020 [26] (https://doi.org/10.1016/j.envexpbot.2020.104262)

Following the reviewer’s suggestion, we have now added information about the years of the experimentation (lines 190-191): Reproducibility was checked by doing a total of three consecutive plant experiments, which were carried out during two years of experimentation (2017-2018).

In the chapter 'conclusions' do not lead the discussions and inquire into the causes of the results - it should be done in the previous chapter.

<Response>: We thank the reviewer for this comment. Conclusions have been rewritten as follows (lines 378-384):

In summary, our results demonstrate the paraquat addition to mineral solution negatively affected soybean growth and development by decreasing plant biomass and nodulation. Also, paraquat altered N2 fixation by reducing SFN by the endosymbiont B. diazoefficiens. Further, paraquat induced an important oxidative stress in plant cells and nodules by increasing lipid peroxidation and reducing Lb content, thus affecting nodule functionality

The results presented may be of minor local importance in countries where paraquat is approved for use.

<Response>: As we mentioned above, our research is not focused on agronomic aspects of using paraquat in soybean crops. We aimed to find new insights about metabolic functionality of nodules under different levels of oxidative stress, with special interest in the role of nodular legheamoglobin.

Considering the above, I do not recommend publishing this manuscript in the journal 'Nitrogen'

<Response>: We are sorry to disappoint this reviewer. In this article, new insights concerning the involvement of legheamoglobin in nodules when they are subjected to moderate and severe oxidative stress are presented and discussed. For the first time, these findings point out that leghaemoglobin is a reliable parameter to assess oxidative stress and hence, SNF in soybean nodules. Therefore, we believed that our results are worthy to share to the scientific community working on symbiotic nitrogen fixation.

Reviewer 3 Report

The research paper, well written . 

I have some  comments:

Line 48: Are the nodules "organ" ? I believe nodules are part of the root with over divided cells, where nitrogen fixing bacteria are accommodated. So it is a root formation ruther than “organ”.

Line 106, section 2.2

The headline says Plant inoculation and growth. That is very well, but the authors, probably need to move here the description of the replications. I think it will be easier to follow.

Line 266: Figure 2.  The authors need mark the values with significant difference (a symbol or letter).

Some sections of materials and Methods need to be rewritten, particular sections 2.2 and 2.4.

Author Response

Dear Editor,

We appreciate very much the comments and suggestions raised by the three reviewers. You will notice that we considered all of them and that almost all of the suggestions were taken into account in the revised version of the manuscript (indicated as track changes mode). Lines, figures, and references in the specific responses correspond to those of the revised manuscript. The text of the each reviewer is repeated here, and our responses are marked as <Response>.

We believe that the quality of the revised manuscript has thereby improved, and we would be happy if it is now acceptable for publication in the Nitrogen journal.

With best regards,

Germán Tortosa and Socorro Mesa

Reviewer 3.

Comments and Suggestions for Authors

The research paper, well written. 

<Response>: Thanks a lot.

I have some comments:

Line 48: Are the nodules "organ" ? I believe nodules are part of the root with over divided cells, where nitrogen fixing bacteria are accommodated. So it is a root formation ruther than “organ”.

<Response>: Thanks for your comment. We agree that nodule is part of the root, but also, according to Stacey (2007, and cites therein), nodule is “a highly specialized organs due to it exhibits cellular specialization”.

We now referred to this article in the Introduction section (line 54) and included it in the reference’s list (lines 426-428):

  1. Stacey, G. Chapter 10 - The Rhizobium-Legume Nitrogen-Fixing Symbiosis, Editor(s): Hermann Bothe, Stuart J. Ferguson, William E. Newton, Biology of the Nitrogen Cycle, Elsevier, 2007, Pages 147-163, ISBN 9780444528575, https://doi.org/10.1016/B978-044452857-5.50011-4.

Line 106, section 2.2

The headline says Plant inoculation and growth. That is very well, but the authors, probably need to move here the description of the replications. I think it will be easier to follow.

<Response>: We thank the reviewer for the good comment. We think that the experimental design and the statistical analysis require a detailed methodological description in which the number of replicates and assays are described. The results showed in our research is a compilation of several experiments done during whole assay, which showed good reproducibility and repeatability.

In order to make our argument easier to follow, we have added the following sentence (lines 138-139):

“A detailed description about experimental procedure including number of replicates is shown in 2.4 section”

Also, we have added more information about experimental design and statistical analysis (lines 193-209):

Plant experiments were designed according to Gomez and Gomez recommendations [33]. We carried out some single-factor experiments based on the randomised block procedure to evaluate the effect of added paraquat to plant physiology and nodular development. Reproducibility was checked by doing a total of three consecutive plant experiments, which were carried out during two years of experimentation (2017-2018). Also, repeatability was assayed by using a total of ten plants (or replicates) per treatment for each plant experiment. Finally, data were expressed as a compilation of all experiments according to their reproducibility and repeatability.

For each experiment and parameter, a descriptive statistical analysis was done, including mean, absolute and relative error calculations. In order to test the effect of paraquat in bacterial and plant development, an inferential statistical analyses based on the Analysis of Variance (ANOVA one-way) with a with post-hoc Tukey test, p<0.05) were calculated. These analyses were done assuming normal distribution and homoscedasticity of the raw data.

The article by Gomez and Gomez has also been included in the references’ list (lines 481-482):

  1. Gomez, K.A.; Gomez, A.A. Statistical procedures for agricultural research. 1984. John Wiley & Sons, Inc.

Line 266: Figure 2.  The authors need mark the values with significant difference (a symbol or letter).

<Response>: Thanks for the suggestions. A new version of Figure 2 with significant differences has been added. The statistical analyses have also been described in the Methods’ section (lines 193-209) as well as in the legend to Figure 2 (line 256).

Some sections of materials and Methods need to be rewritten, particular sections 2.2 and 2.4.

<Response>: As mention above and following the suggestion of this reviewer, we have added extra information in Sections 2.2 (Lines 138-139) and 2.4 (Lines 193-209)

Round 2

Reviewer 2 Report

Dear Authors

After analyzing the authors' explanations and the new version of the manuscript, I recommend it for publication in the Nitrogen journal

Reviewer